# Earth Worming—An Evaluation of Earthworm (*Eisenia andrei*) as an Alternative Food Source

**DOI:** 10.3390/foods12101948

**Published:** 2023-05-10

**Authors:** Ruchita Rao Kavle, Patrick James Nolan, Alan Carne, Dominic Agyei, James David Morton, Alaa El-Din Ahmed Bekhit

**Affiliations:** 1Department of Food Science, University of Otago, Dunedin 9054, New Zealand; kavru760@student.otago.ac.nz (R.R.K.); patricknolan28@outlook.com (P.J.N.); dominic.agyei@otago.ac.nz (D.A.); 2Department of Biochemistry, University of Otago, Dunedin 9054, New Zealand; alan.carne@otago.ac.nz; 3Department of Wine, Food and Molecular Biosciences, Lincoln University, Christchurch 7647, New Zealand; jim.morton@lincoln.ac.nz

**Keywords:** *Eisenia andrei*, proximate composition, lipid nutritional indices, mineral profile, fatty acid profile, amino acid profile, essential amino acid index, isoelectric point, foam capacity

## Abstract

Aside from their bioremediation roles, little is known about the food and feed value of earthworms. In this study, a comprehensive evaluation of the nutritional composition (proximate analysis and profiles of fatty acids and minerals) and techno-functional properties (foaming and emulsion stability and capacity) of earthworm (*Eisenia andrei*, sourced in New Zealand) powder (EAP) were investigated. Lipid nutritional indices, ω6/ω3, atherogenicity index, thrombogenicity index, hypocholesterolemic/hypercholesterolemic acid ratio, and health-promoting index of EAP lipids are also reported. The protein, fat, and carbohydrate contents of EAP were found to be 53.75%, 19.30%, and 23.26% DW, respectively. The mineral profile obtained for the EAP consisted of 11 essential minerals, 23 non-essential minerals, and 4 heavy metals. The most abundant essential minerals were potassium (8220 mg·kg^−1^ DW), phosphorus (8220 mg·kg^−1^ DW), magnesium (744.7 mg·kg^−1^ DW), calcium (2396.7 mg·kg^−1^ DW), iron (244.7 mg·kg^−1^ DW), and manganese (25.6 mg·kg^−1^ DW). Toxic metals such as vanadium (0.2 mg·kg^−1^ DW), lead (0.2 mg·kg^−1^ DW), cadmium (2.2 mg·kg^−1^ DW), and arsenic (2.3 mg·kg^−1^ DW) were found in EAP, which pose safety considerations. Lauric acid (20.3% FA), myristoleic acid (11.20% FA), and linoleic acid (7.96% FA) were the most abundant saturated, monounsaturated, and polyunsaturated fatty acids, respectively. The lipid nutritional indices, such as IT and ω-6/ω-3, of *E. andrei* were within limits considered to enhance human health. A protein extract derived from EAP (EAPPE), obtained by alkaline solubilisation and pH precipitation, exhibited an isoelectric pH of ~5. The total essential amino acid content and essential amino acid index of EAPPE were 373.3 mg·g^−1^ and 1.36 mg·g^−1^ protein, respectively. Techno-functional analysis of EAPPE indicated a high foaming capacity (83.3%) and emulsion stability (88.8% after 60 min). Heat coagulation of EAPPE was greater at pH 7.0 (12.6%) compared with pH 5.0 (4.83%), corroborating the pH-solubility profile and relatively high surface hydrophobicity (1061.0). These findings demonstrate the potential of EAP and EAPPE as nutrient-rich and functional ingredients suitable as alternative food and feed material. The presence of heavy metals, however, should be carefully considered.

## 1. Introduction

The ability to attain global food and nutrition security is becoming substantially challenged by world population growth and hampered by climate change which is a major global problem today. Climate change and its effects, such as droughts, floods, storms, and extreme temperatures, have contributed substantially to the growth in world hunger [1]. Earthworm species such as *Eisenia foetida, Eisenia andrei, Dendrobaena veneta, Hyperiodrilus euryaulos, Lumbricus terresstris,* and *Pseudoneoponera excavates* have been widely used in the animal feed sector, as a source of protein for fish, chickens, rabbits, and pigs [2,3,4]. 

As earthworms have a substantial protein content [5,6,7], earthworm protein is therefore being utilised as an ingredient in pelleted animal feeds in the United States, Canada, and Japan [8]. On a dry weight basis, earthworms are typically composed of 65% protein, 14% fat, 14% carbohydrate, and 3% ash [9,10,11,12,13]. It is reported that feed meals produced from earthworms can typically have a higher protein content and a nutritionally better amino acid composition than meals made from fish or soybeans [14]. Despite this high nutrient composition, earthworms are not widely accepted as a human food source. 

There is an emerging trend of a search for “superfood materials”, and hence earthworm biomass has nutritional appeal. In the 1970s, earthworm-containing dishes were already well known [15], and in some countries, earthworm material is utilised as a nutritional supplement [16], with unique cuisines based on earthworms still being created in the provinces of Taiwan, and in Henan and Guangdong, China. 

Earthworms from the family Glossoscolecidae are a major food source for the Ye’Kuana Amerindian tribe in South America [14,17,18] due to their high protein content. Consequently, it is reported that there is an average annual consumption of 1.7–2 kg of *Andiorrhinus kuru* and *Andiorrhinus moto* in Venezuela [17].

The value of earthworms as a supplement in animal dietary formulations and as human food has been established, and therefore there is a need to determine the chemical composition and fatty acid profile of *E. andrei*, which could guide the use of this species in feed and food formulations. A previous study in Brazil by Rodrigues et al. [19] focused on the enzymatic hydrolysis of *E. *andrei** and characterized and evaluated the techno-functional properties of the earthworm. There has been no study reporting an in-depth analysis of the nutrient content and techno-functional properties of protein from *E. andrei* reared in New Zealand, hence the study reported here. 

## 2. Materials and Methods

### 2.1. Sample Collection and Proximate Composition of E. andrei Powder (EAP)

Live earthworms were obtained from Biosuppliers Live Insects, Dunedin, New Zealand (http://biosuppliers.nz/ accessed on 16 November 2020). *E. andrei* (see Figure 1A) were farmed in soil and were fed various vegetable kitchen scraps and eggshells, and harvested after approximately 50 days. The harvested products were packaged in breathable boxes and then delivered to our research laboratory in Dunedin, New Zealand, arriving within a week from dispatch due to logistics during the COVID-19 pandemic. Upon arrival, the worms, washed in water, were starved for 24 h, after which they were re-washed, frozen, and freeze-dried for 48 h using a LABCONCO freeze-drier (FreeZone 12 Plus, KCMO, Kansas City, MO, USA). The freeze-dried samples were ground and stored at −20 °C until further analysis. The dried powder is referred to as *E. andrei* powder (EAP). 

The moisture (AOAC 950.46) and ash (AOAC 920.153) contents were determined by proximate composition analysis in accordance with the Association of Official Analytical Chemists [20]. According to AOAC [20], the Kjeldahl and Soxhlet techniques were used to determine the total crude protein (AOAC 981.10) and lipid (AOAC 960.69) contents, respectively. A conversion factor of 6.25 was used to convert the total nitrogen data from the Kjeldahl technique to crude protein [21]. Equation (1) was used to calculate the carbohydrate fraction (%).
Carbohydrate (%) on dry weight basis (DW) = 100 − sum (%crude protein DW + %ash DW + %lipid DW)(1)

The total energy available in the samples was calculated from the complete crude protein, lipid, and carbohydrate contents using Equation (2), as reported by FAO UN (2003).
Energy(kJ/100 g) = (total crude protein × 17) + (lipid × 37) + (carbohydrate × 17)(2)

### 2.2. Fatty Acid Analysis of EAP

The fatty acid content was analysed as fatty acid methyl esters (FAMEs) as reported in Kavle et al. [22], and the values were expressed as % fatty acid, and g fatty acid/100 g of dried EAP material (Table 1).

### 2.3. Dietary Indicators of EAP

Lipid dietary indicators, comprising the hypocholesterolemic/hypercholesterolemic ratio, the index of atherogenicity (IA), the index of thrombogenicity (IT), and the health-promoting index (HPI), were determined using Equations (3)–(6) [23]:IA = [12: 0 + (4 × 14:0) + 16:0]/ΣUFA(3)
IT = (14:0 + 16:0 + 18:0)/[(0.5 × ΣMUFA) + (0.5 × Σω6 PUFA) + (3 × Σω3PUFA) + (ω3/ω6)](4)
HH = (18:1 cω9 + ΣPUFA)/(12:0 + 14:0 + 16:0)(5)
HPI = ΣUFA/[12:0 + (4 × 14:0) + 16:0](6)

ΣPUFA (polyunsaturated fatty acid) = 18:2 cω6 + 18:3 ω3

ΣUFA (unsaturated fatty acid) = ΣMUFA + ΣPUFA

ΣMUFA (monounsaturated fatty acid) = 16:1 cω7 + 18:1 cω9

Σ = Sum of the fatty acid components

ω = Omega, which refers to the first double bond from the methyl end of the fatty acid

### 2.4. Mineral Analysis of EAP

Mineral analyses were performed as reported by Kavle et al. [22]. In a Class 10 (ISO4) clean and metal-free laboratory, aliquots (0.25 g) of EAP were investigated for mineral content using an inductively coupled plasma mass spectrometer (Agilent 7850, Santa Clara, CA, USA; Department of Chemistry, University of Otago, New Zealand). 

In addition to using the EAP for quality control, a reference material fish protein certified substance (DORM-4, NRC-CNRC) was also used.

### 2.5. Protein Extraction Yield of EAP

Earthworm (*E. andrei*) total protein was recovered from EAP using alkaline extraction and isoelectric pH precipitation following freeze drying (LABCONCO, FreeZone 12 Plus, KCMO, Kansas City, MO, USA) for 48 h, with a slight modification of the method described by Kavle et al. [22].

Equation (7) was used to calculate the yield of protein extracted from the EAP.
(7)$$\mathrm{Protein}\;\mathrm{extract}\;\mathrm{yield}\;\left(\%\right)=(\mathrm{Weight}\;\mathrm{of}\;\mathrm{the}\;\mathrm{dried}\;\mathrm{protein}\;\mathrm{extract}\;\mathrm{powder}\;\mathrm{recovered}/\;\mathrm{Total}\;\mathrm{weight}\;\mathrm{of}\;\mathrm{the}\;\mathrm{dried}\;\mathrm{insect}\;\mathrm{material}\;\mathrm{used})\times 100$$

The EAP protein extract is referred to as EAPPE in subsequent sections. 

### 2.6. Amino Acid Composition of EAPPE

The amino acid composition of EAPPE was determined using the method of Jayawardena et al. [24] using an Agilent 1100 series HPLC (Agilent Technologies, Waldbronn, Germany). 

### 2.7. Determination of the Essential Amino Acid Index (EAAI) of EAPPE 

The essential amino acid index (EAAI), which is based on the amount of all essential amino acids in comparison with a reference protein, was used to assess protein quality. According to Yang et al. [25], the EAAI gives a measure of the protein quality of a product for human consumption and was computed using Equation (8).
(8)$$=\sqrt[{n}]{\frac{mg\;of\;lysine\;in\;1\;g\;of\;E.\;andrei\;}{mg\;of\;lysine\;in\;1\;g\;WHO\;daily\;human\;\mathrm{adult}\;\mathrm{requirement}}\times \left(\ldots other\;essential\;amino\;acid\right)}$$

The FAO/WHO [26] recommended mg·g^−1^ daily adult human requirement for essential amino acids was chosen as the reference protein in the current study.

### 2.8. Colour Measurement of EAPPE

The colour of the *E. andrei* dried protein extracts was measured using a Hunterlab miniscan XE Spectrocolorimeter (Hunterlab, Reston, VA, USA). The lightness (*L**), red/green (*a**), and blue/yellow (*b**) values were determined in triplicate. The colorimeter was calibrated using white and black reference tiles before measuring the samples.

Equations (9) and (10) were used to calculate the browning index (BI).
BI = [100 × (x − 0.31)]/0.17(9)
x = (*a** + 1.75 × *L**)/(5.645 × *L** + *a** − 3.012 × *b**)(10)

### 2.9. Particle Size Analysis of EAPPE Suspension

A 0.45 μm pore size syringe filter was used to filter a suspension of EAPPE powder (1% *w*/*v*) that had been suspended in 5 mL of 2 mM sodium phosphate, pH 7.0, for 1 h (Merck, Darmstadt, Germany). The particle size distribution in the protein solution was assessed in triplicate, using a Nano Series Zetasizer (Malvern, Australia), according to Queiroz et al. [27]. Measurements were made at a temperature of 25 °C and a scattering angle of 173°.

### 2.10. Surface Hydrophobicity of EAPPE Suspension 

The protein surface hydrophobicity of EAPPE was determined [28], with slight modification. A Synergy 2 Microplate Reader (BioTek Instruments, Inc., Hudson, MA, USA) was used to measure the fluorescence intensity (excitation 390 nm, emission 470 nm). The surface hydrophobicity value was calculated using the slope of a linear regression plot of the protein content and fluorescence intensity.

### 2.11. Differential Scanning Calorimetry (DSC) of EAPPE

The thermal stability of the defatted EAPPE was determined according to Queiroz et al. [27] using differential scanning calorimetry (DSC) analysis. Measurements were made in triplicate of 4 mg aliquots of EAPPE using a DSC 250 instrument (T.A. Instruments Ltd., New Castle, DE, USA). The samples were heated over the range 10 °C to 250 °C, at a heating rate of 5 °C/min. The T.A. Universal Analysis 200 programme (T.A. Instruments Ltd., New Castle, DE, USA) was used to calculate the temperatures for the start (T_o_), peak (T_p_), and end of the denaturation of both protein extracts (Tc), change in enthalpy (H, determined by integrating the area under the endothermic peak), and temperature range (∆T_d_ = T_c_ − T_o_).

### 2.12. Fourier-Transform Infrared Spectroscopy (FTIR) of EAPPE

The FTIR spectra of the defatted EAPPE was determined according to the technique of Queiroz et al. [27] using a Bruker Optics Fourier-transform infrared (FTIR) spectrometer (Alpha Systems, Waltham, MA, USA) equipped with an attenuated total reflection (ATR) platinum diamond 1 accessory. 

### 2.13. Foaming Capacity and Stability of EAPPE Suspension

EAPPE suspension foaming capacity and foam stability were determined [28], with slight modifications. 

Foam layer volume was measured at 5 min, 10 min, 15 min, 30 min, 60 min, 90 min, and 120 min after homogenization to gauge the stability of the foam. Equation (12) was then used to compute the foam stability, where FC is the foaming capacity and FC_0_ is the foaming capacity at time zero.
(11)$$\mathrm{Foaming}\;\mathrm{capacity},\;\%=\frac{H_{t}-H_{0}}{H_{0}}\times 100$$
(12)$$\mathrm{Foaming}\;\mathrm{stability},\;\%=\frac{\mathrm{FC}}{{\mathrm{FC}}_{0}}\times 100$$

### 2.14. Water- and Oil-Holding Capacity of EAPPE Suspension

The methodology of Clarkson et al. [28] was used to measure the water- and oil-holding capacity (WHC, OHC). Decanting the supernatant was carried out with care. Equation (13) was used to determine the ability to contain oil or water, expressed as mL·g^−1^.
(13)$$\mathrm{Water}-\;\mathrm{or}\;\mathrm{oil}-\mathrm{holding}\;\mathrm{capacity}\;(\frac{\mathrm{mL}}{g})=\frac{\mathrm{water}\;\mathrm{or}\;\mathrm{oil}\;\mathrm{held}\;\left(\mathrm{mL}\right)}{\mathrm{Sample}\;\mathrm{weight}\;\left(g\right)}$$

### 2.15. Emulsifying Activity/Capacity of EAPPE Suspension

Determination of emulsifying capacity (EC) and emulsion stability (ES) of the EAPPE suspensions were based on Kim et al. [1]. The emulsifying capacity was calculated using Equation (14), where H_f_ is the initial sample volume (mL) and H_i_ is the sample volume after homogenisation. After homogenisation, 50 μL of the emulsion was immediately mixed with 10 mL of 0.3% (*w*/*v*) SDS solution to assess the stability of the emulsion.

After gentle inversion, the light absorbance of the mixture at 500 nm was measured at intervals of 10 min, 20 min, 30 min, 60 min, 90 min, and 120 min. Equation (15), where E_t_ is the absorbance at different time intervals and E_0_ is the absorbance of the emulsion at 500 nm at time zero, was used to compute the emulsion stability.
(14)$$E\mathrm{mulsifying}\;\mathrm{capacity}\;\left(\mathrm{EC}\right),\;\%=\frac{H_{f}}{H_{i}}\times 100$$
(15)$$\mathrm{Emulsion}\;\mathrm{stability}\;\left(\mathrm{ES}\right),\;\%=\frac{E_{0-}E_{t}}{E_{0}}\times 100$$

### 2.16. Solubility Profile of EAPPE Suspension

The solubility was measured based on the method of Mishyna et al. [29], with slight modification. The protein concentration of the supernatant was determined using the Lowry assay [30]. Calculation of protein solubility involved comparing the amount of supernatant soluble protein to total amount of protein in the EAPPE sample.

### 2.17. Coagulation of EAPPE

The heat-induced coagulation of EAPPE suspensions was performed as reported by Mishyna et al. [29], with slight modification. Equation (16) was used to compute the protein coagulation (%), where *A*_0_ represents the absorbance at 540 nm prior to heating, and *A_t_* represents the absorbance following heating.
(16)$$\mathrm{Coagulation},\;\%=\frac{A_{0-}\;A_{t}}{A_{0}}\times 100$$

### 2.18. Statistical Analysis

The data obtained from three replicate independent experiments were tabulated using Microsoft Excel, and the results are reported as mean ± standard deviation (SD). A graphical representation of the means was generated using the same software. 

## 3. Results and Discussion 

### 3.1. Proximate Analysis of EAP

To date, *E. andrei* is one of the least studied species of earthworm. To the best of our knowledge, this is the first report on the nutrient composition of *E. andrei*. In the present study, the proximate composition of EAP, on a dry weight basis, was as follows: protein 53.75%; fat 19.30%; ash 3.69%; and carbohydrate 23.26%. The energy content was determined to be 2023.2 kJ/100 g. The moisture content of fresh earthworm material was determined to be 83.68%. The moisture content of fresh *E. andrei* is higher than that reported for Honingka worm, *Siphonosoma australe* (79.59%) [11]. Freeze drying improves the stability of stored earthworm material and preserves some sensory attributes and heat-labile nutrients, which results in improved performance of earthworm material in food and feed [31]. The ash content of the freeze-dried *E. andrei* powder (EAP) was found to be 3.69% (DW basis), which is higher than that reported for E. foetida (1.4% *w*/*w*) [32], but comparable to the Honingka worm (3.2% DW) that is a traditional food of the ethnic people of Sombu Coastal community of Wakatobi Islands, Southeast Sulawesi, Indonesia [11]. 

The fat content of EAP in the present study (19.30% DW) was higher than that of the Honingka worm (6.35% DW) [11], *Perionyx excavates* (7.8% DW) [33], *E. foetida* (10.0% DW) [34], and *Dendrobaena veneta* (12.21% DW) [13]. As observed, the fat content of worm meal has been reported to range from 5–20% of dry matter [35]. In addition to their calorie input, fats improve the palatability of food and supply essential fatty acids. The metabolism of lipids can change during different stages of worm development and differences in the fat content in different organisms can be influenced by habitat location, nutrition, and how the worms are reared [36]. 

The total crude protein content of EAP was found to be 53.75% DW. This is lower than that for *E. foetida* [37], for which crude protein values of 66.2% and 59.7% DW for frozen and oven-dried samples, respectively, were reported. Work by Rondón et al. [10] on *E. foetida*, reported crude protein contents of 62.28% and 61.81% DW after freezing and oven drying, respectively. Hence, freeze drying is a slightly more effective method compared with oven drying and sun drying for retaining the protein content of the worm material during drying [31,34]. 

The total protein content found in the present study of *E. andrei* was lower than that reported for other earthworm species, such as *D. veneta* (76.82% DW) [13], *Hyperiodrilus euryaulos* (63.0% DW) [37], and *L. terresstris* (64.0% DW) [38]. The relatively high levels of protein reported for Honingka worms (86.94% DW) from a previous study [11] are thought to be due to a large amount of sediment deposited in the Honingka worm habitat. In other studies by Rodrigues et al. [19], a higher protein content (70.9%) was obtained when compared with that found in the present study. The nutritional content of the worms may vary depending on the diet, ecology, geographic location, and harvest location, and the processing methods (e.g., mechanical and thermal processing techniques) may affect the nutrient content of the material [38]. 

### 3.2. Fatty Acid Composition of EAP

The profile of fatty acids (FAs) in EAP is shown in Table 1 (% total FA, g FA per 100 g DW). The most abundant saturated fatty acid (SFA) was found to be lauric acid (C12:0) (20.3% of total FA), followed by palmitic acid (12.8% of total FA), and stearic acid (C18:0) (5.03% of total FA). The World Health Organization (WHO) has advised against consuming elevated levels of SFA because of their link to raised levels of low-density lipoprotein cholesterol (LDL-C) and a higher prevalence of heart disease [39]. While there are many recommendations to reduce the overall consumption of saturated fatty acids (SFAs), it is important to consider the different health impacts of specific SFAs in the diet. [40]. For example, in oils such as coconut and palm oil, lauric acid (LA), a medium-chain saturated FA is present in high amounts and has been reported to have a cholesterol-raising effect, increasing both total cholesterol levels and low-density lipoprotein (LDL) cholesterol [41]. However, it is also reported that LA increases the levels of high-density lipoprotein cholesterol (HDL-C) and may decrease the total cholesterol to HDL-C ratio [16], both of which are beneficial for cardiovascular health [42]. Hence, the cardiovascular effect of LA on dyslipidemia may be beneficial, and at worst benign rather than harmful [40]. 

Owing to the low amounts of myristic acid in EAP (1.05% of FA, Table 1), which is known to contribute to hypercholesterolemia when present at high levels [39], it can therefore be inferred that ingesting EAP would have beneficial health benefits.

The most prevalent monounsaturated fatty acids (MUFA) identified in EAP were myristoleic acid (11.20% of total FA), palmitoleic acid (16:1 cω7) (6.94% of total FA), and oleic acid (18:1 cω9) (4.19% of total FA) (Table 1). The total MUFA was lower than the SFA and PUFA contents of EAP (Table 1). This trend is similar to that reported in freeze-dried *E. foetida* (SFA 48.1% of total FA; MUFA 23.4% of total FA; PUFA 28.5% of total FA) [34]. Oleic acid, which was found in the earthworms examined in this study, is known to contribute to reducing blood pressure and the risk of developing inflammatory, autoimmune, and cardiovascular disorders when consumed by humans [43]. 

In the present study, it was found that 28.37% of the total FA in EAP was made up of polyunsaturated fatty acid (PUFA). One of the most abundant FAs in EAP (Table 1) was found to be linoleic acid (7.96% FA) and the UFA content (54.9% of total FA) was higher than the SFA content (44.45% of total FA). However, worms such as *L. rubellus* are reported to have higher levels of SFA (37.75% of total FA), MUFA (44.94% of total FA), and PUFA (16.21% of total FA) [44], and *D. veneta* is reported to have a SFA content of 33.38% of total FA, MUFA (39.67% of total FA), and PUFA (14.62% of total FA) [13]. Hence, the findings obtained for EAP in the present study are in general agreement with those in the published literature, and the small variations could be attributed to factors such as differences in feed and stage of development of the worm. 

### 3.3. Nutritional Indices

#### 3.3.1. ω-6/ω-3 of EAP

In the current investigation, the fatty acid ω-6/ω-3 ratio for the lipid extracted from EAP was found to be 4.11 (Table 1), which was within the <10 value proposed by WHO, as reported by Kumar et al. [45]. In other studies, the ω-6/ω-3 ratio for *E. *foetida** was 1.97 [34], for *L. rubellus* it was 1.99 [44], and for *L. rubellus* it was 1.74 [46], which were all lower than that observed for *E. andrei* in the present study. However, the *D. veneta* ω-6/ω-3 (7.85) and *E. foetida* (8.72) [13] are higher than that reported for EAP in the present study. The difference in the ω-6/ω-3 ratio in insects and invertebrates has been reported to be dependent on the diet [47]. A lower ratio of ω-6/ω-3 is thought to assist in lowering chronic disease presentation (diabetes, autoimmune disease, and cardiovascular disease) [48]. 

#### 3.3.2. PUFA/SFA Ratio of EAP 

The ratio of PUFA/SFA is a primary criterium typically used to assess lipid nutritional quality in food. As previously stated by WHO [49], it is considered that the PUFA/SFA ratio should ideally be more than 0.4 in order to satisfy dietary requirements. A high ratio is thought to contribute to positive cardiovascular health [50]. The ratio of PUFA/SFA in EAP was found to be 0.23 (Table 1), which is lower than that reported for *E. foetida* (1.2) [34], *L. rubellus* (0.4) [46], and *D. veneta* (0.43) [13]. Although the value obtained for EAP in the present study does not meet the considered ideal PUFA/SFA ratio, it is better than the PUFA/SFA ratio reported for palm stearin (0.13), *Loxechinus albus* (shellfish) (0.20), and dairy products (0.02–0.175) [50].

#### 3.3.3. Index of Atherogenicity (IA) of EAP

Ulbricht et al. [23] defined the index of atherogenicity (IA) as a relationship of saturated fatty acids (SFAs) and unsaturated fatty acids (UFAs), as well as the potential of FA in diet atherogenic capability. UFA is considered to be anti-atherogenic since it has been reported to reduce plaque buildup in the circulatory system and aid in cholesterol level reduction, phospholipids, and esterified fatty acids [51,52]. The IA value for EAP was determined to be 1.32, which is higher than that reported for freeze-dried *E. foetida* (0.49) [34], *L. rubellus* (0.78) [46] and *D. veneta* (0.75) [13], and animal meat (typically ranging from 0.5 for chicken, foal, and bologna sausages to 1.0 for rabbit, lamb, and heifer) [50].

#### 3.3.4. Index of Thrombogenicity (IT) of EAP

The index of thrombogenicity (IT) provides an indication of the potential of blood arteries to produce clots, which provides a consideration of the link between pro-thrombogenic (saturated) and anti-thrombogenic fatty acids (MUFAs, PUFAs—ω6 and PUFAs—ω3) [43]. In the present study, the IT value for EAP was found to be 0.55, which is lower than that previously reported for freeze-dried *E. foetida* (0.64) [34] and certain edible insects, such as *Acheta domestica* (1.25) [35], locust (1.51) [53], *Rhynchophorus ferrugineus* (1.35) [54], and that reported for beef (0.70) [55]. Even though no precise IA or IT values have been advised internationally, it is believed that a low IT value indicates a higher quality nutritionally and a lower risk of developing coronary heart diseases. Therefore, based on these two indices, EAP could be considered to be of good nutritional quality in comparison with other sources of food.

#### 3.3.5. Hypocholesterolemic/Hypercholesterolemic (HH) Value of EAP

With regard to systemic cholesterol levels and the impact of FAs, a measure called the hypocholesterolemic/hypercholesterolemic (HH) ratio is utilised. It is recognised that HH values provide a more precise measure in relation to the effect of FA on the risk of cardiovascular illnesses than the PUFA/SFA ratio [56]. A higher HH ratio is considered to be better for human health [50]. The HH ratio for EAP was determined as 0.44, which is lower than that of *E. foetida* (1.61) [34] and *D. veneta* (1.87) [13]; hence by this measure, EAP might be considered to be of lower health-promoting value, based on the HH ratio, compared with other species of earthworm. 

#### 3.3.6. Health-Promoting Index (HPI) of EAP 

A higher value of the health-promoting index (HPI), a measure that was proposed in previous literature [57] to measure the nutritional value of dietary fat, is considered to be a useful indicator of the contribution to human health. The HPI (which is, in effect, the opposite of the IA) was determined to be 0.78 for EAP in the present study, which is higher than for *D. veneta* (0.39) [13] and for dairy products (0.16 to 0.68) [58,59].

### 3.4. ICP-MS Analysis of Minerals in EAP

The mineral content (mg·kg^−1^ DW) of *E. andrei* was investigated in the present study using ICP-MS, and a reference material (fish protein, DORM-4) was used for quality control (Table 2). A total of 5 macrominerals, 6 microminerals, and 4 heavy metals were quantified and the results are presented in Table 2. 

#### 3.4.1. Essential Macrominerals of EAP 

In the present study, potassium (K) was found to be the most abundant mineral in freeze-dried earthworms, with an outstanding average of 8220 mg·kg^−1^ DW. This is within the range reported in previous studies in which K was reported to be the most abundant mineral in *L. mauritii* powder (3850 mg·kg^−1^ DW) [60], in a defatted meal of *E. foetida* (4723 mg·kg^−1^ DW) [61], in the body fluid of *E. foetida* (990 mg.l^−1^) [32], and in *L. terresstris* (11,030 mg·kg^−1^ DW) [62]. The K content of EAP found in the present study is comparable to that reported for several invertebrates such as *L. terresstris* [62] but is higher than that reported for the majority of edible insects (superworms, giant mealworms, mealworms, waxworms, silkworms, and crickets) [48].

The other relatively abundant macrominerals found in the EAP in the present study were phosphorus (6600), sodium (4056.7), calcium (2396.7), and magnesium (744.7) (all values in mg. kg^−1^ DW) (Table 2). The phosphorus content of EAP found in the present study is less than that reported for several other worms and edible insects such as mealworms, silkworms, the earthworm *L. terresstris*, and crickets (ranging from 7480 to 13,700 mg·kg^−1^ DW), except for waxworms and superworms (4699 mg·kg^−1^ and 5629 mg·kg^−1^ DW) [61]. In the present study, the phosphorus content found for EAP is either lower (e.g., *Tenebrio molitor, Acheta domesticus*, *Bombyx mori*, *Apis mellifera*, and *Schistocerca gregaria*), similar (*Gryllus bimaculatus*, *Locusta migratoria*, and *Rhynchophorus phoenicis*), or higher (*Hermetia illucens* and *Imbrasia oyemensis*) than that reported for these edible insects [48]. The phosphorus content in EAP is higher than that reported for traditional protein sources, such as raw chicken, whole fresh egg, raw beef, and soybean (1740 mg·kg^−1^, 198 mg·kg^−1^, 1820 mg·kg^−1^, and 1940 mg·kg^−1^, respectively) [63].

The sodium (4056.7 mg·kg^−1^ DW) and calcium (2396.7 mg·kg^−1^ DW) contents of EAP found in the present study are lower than that reported for *Lumbricus terresstris* (5524 mg·kg^−1^ and 2707 mg·kg^−1^ DW, respectively) [62]. The magnesium content of EAP in the present study (744.7 mg·kg^−1^) is lower than that reported for superworms, giant mealworm larvae, mealworm larvae, waxworms, and silkworms (range 316 mg·kg^−1^ to 864 mg·kg^−1^ frozen, and moisture contents in the range of 57.9% to 82.7%). The edible insects *Bombyx mori* and *Apis mellifera* are reported to have higher Mg content (2000–2900 mg·kg^−1^ DW) [48] and *Prionoplus reticularis* (1306.7 mg·kg^−1^ DW) [63] than that reported for EAP in the present study. Conversely, the magnesium content of EAP is higher compared with *L. mauritii* (385 mg·kg^−1^ DW) [60] and *L. terresstris* (136 mg·kg^−1^ frozen and 83.6% moisture content) [62].

Macrominerals have important physiological roles. For example, potassium is involved in neurological and cellular functions [64], and a deficiency has been reported to be implicated in diabetes and weight gain [65]. Phosphorus and calcium are important for the formation of teeth and bones [66] and several biological processes. The recommended daily intakes of calcium (1000–1300 mg), magnesium (280–350 mg), potassium (2300–3000 mg), sodium (>500 mg and <2300 mg) (Table 2), and phosphorus (500–1250 mg) [67] indicate that 100 g of EAP could provide about 20% of the recommended daily intakes (RDIs) of sodium and calcium and 40% of the RDIs of potassium, magnesium, and phosphorus.

#### 3.4.2. Essential Microminerals of EAP

The iron content in EAP was found to be 244.7 mg·kg^−1^ DW (Table 2). Iron serves as a micronutrient and is essential for the body’s ability to carry oxygen in the blood and for some essential enzymes in tissues [68]. EAP can be considered a potential source of iron as it meets the recommended dietary allowance (RDA) [69]. Children, teenagers, and women of reproductive age are the groups most at risk of presenting with iron deficiency [70]. *L. mauritii* powder is reported to contain a level of iron (241.1 mg kg^−1^ DW) [60] comparable to that of EAP. Levels of manganese in EAP were found to be 25.6 mg·kg^−1^ DW, slightly higher than that of *L. mauritii* powder (17.2 mg·kg^−1^ DW) [60]. Manganese is a cofactor of several vital enzymes, including pyruvate carboxylase and superoxide dismutase, and it aids in the growth of bone and cartilage, as well as the healing of wounds [71]. Nevertheless, high manganese intake is reported to be harmful [71]. A higher level of zinc was found in EAP compared with that of *L. mauritii* powder (32.34 mg·kg-1) and comparable to some edible insects such as Bombay locust (82.2 mg·kg^−1^ DW) and scarab beetle (88 mg·kg^−1^ DW) [72]. New Zealand soils are reported to contain a comparatively low level of zinc [73]. According to FAO/WHO [68], zinc contributes to the function of over 300 enzymes involved in the metabolism of micro- and macro-nutrients, supports cellular and organ integrity, and affects gene expression and immunity. 

Selenium (Se) was found to be detectable in EAP (0.6 mg·kg^−1^ DW). The trace element selenium is important, and a lack of Se has been linked to a number of illnesses, including immune system dysfunction [74]. Se is required for antioxidant enzymes. However, it is reported that increased Se consumption can raise the risk of type 2 diabetes [75]. Chromium is another essential micronutrient that was detected by ICP-MS in EAP (0.2 mg·kg^−1^ DW) and is essential for humans. Its primary purpose is to control blood sugar levels, and chromium deficiency has been reported to be linked to coronary heart disease, hypoglycemia, food allergies, and symptoms similar to diabetes [76]. 

Therefore, EAP consumption could mitigate the risk of zinc, iron, and manganese deficiency. 

#### 3.4.3. Non-Essential Minerals—Heavy Metals

Consuming worms that have been wild harvested raises questions as to whether heavy metals have been accumulated from the environment. In the present study, heavy metals such as arsenic, vanadium, lead, and cadmium were found to be present above the detectable limit in EAP (Table 2).

Levels of cadmium (2.2 mg·kg^−1^ DW) and arsenic (2.3 mg·kg^−1^ DW) were found to be present in EAP, which are considered to be relatively high levels. The level of cadmium in New Zealand soil is reported to be 0.18 to 0.89 mg·kg^−1^ [77]. For leafy vegetables, cereal, root, and tuber crops, Food Standards Australia New Zealand (FSANZ) has established a recommended level of cadmium for consumer safety of <0.1 mg·kg^−1^ fresh weight [78]. It is recognized that the use of phosphate-based fertilisers in agriculture in New Zealand can result in a modest increase in accumulated cadmium levels. Arsenic has been found in all earthworm species studied at concentrations ranging from 0.4 to 53.6 mg·kg^−1^ DW [79]. Arsenic, in regard to ecological safety, has become a worldwide public concern, as human exposure to arsenic is mostly through water and food [79]. As the source of accumulation of arsenic is through the soil, it is challenging to control the accumulation of heavy metals such as arsenic in wild-harvested earthworms. However, farming earthworms fed with food waste that has been checked for arsenic levels could provide a means to control arsenic accumulation. It is also recognized that although earthworms can accumulate harmful substances, the extent depends on the food and soil characteristics [80]. Tomlin [81] reported that the ecological traits of different earthworm species, such as whether they are endogeic, anecic, or epigeic, may alter how susceptible they are to the accumulation of soil-borne toxins. Although aerial deposition of heavy metals from, for example, smelters, may have an impact, the typically shallow soil depth dwelling lifestyle of earthworms may result in reduced exposure risk compared with deeper burrowing animals, both through cutaneous and dietary uptake [82].

### 3.5. EAPPE Yield and Protein Recovery 

The yield of freeze-dried earthworm alkaline extract protein was found to be 22.1% DW (Appendix A) in the present study. These values are lower than that of some edible insects such as *A. mellifera* (27.5% DW) [29], *Rhynchophorus ferrugineus* (65.7% DW) [83]. Moreover, the protein recovery in EAPPE was found to be 78.7% DW. The total protein extracted from *E. andrei* was found to be greater than 65% on the dry weight basis, hence EAPPE can be considered to be a ‘protein extract’. 

### 3.6. Amino Acid Profile of EAPPE 

The EAPPE amino acid composition is provided in Table 3. Around 19 amino acids were detected in EAPPE, including 8 amino acids that are essential. The total essential amino acids (TEAA) content was 373.3 mg·g^−1^ protein for EAPPE, which is higher than that reported for earthworm species (*E. foetida* (258.8 mg·g^−1^ protein) [84], *E. foetida* (202.9 mg·g^−1^ protein) [2], and *E. foetida* (346.0 mg·g^−1^ protein) [9]). Moreover, the TEAA value of EAPPE was higher than that reported for some insects, such as black solider fly larvae (329. 3 mg·g^−1^ protein) [85], wasp larvae (305.5 mg·g^−1^ protein) [85], and some conventional protein sources, such as whey (341 mg·g^−1^ protein) [86], but lower than chickpea (420.4 mg·g^−1^ protein) [87]. 

The essential amino acids present at the highest concentration in EAPPE were leucine (81.1 mg·g^−1^ protein) followed by lysine (71.2 mg·g^−1^ protein). Leucine plays multiple roles in metabolism beyond the requirement as an amino acid in protein synthesis [88]. Lysine is a limiting amino acid in the diet of many people around the world who rely on cereals as a staple in their diet [89] and is an essential AA. 

The histidine content of EAPPE (27.9 mg·g^−1^ protein) is above the recommended value of 15 mg·g^−1^ protein [90]. For the production of numerous hormones crucial for metabolic processes, histidine is a vital building block. However, it has been observed that consuming foods containing elevated histidine, which can form histamine, might cause allergic responses in some individuals [91]. Interestingly, the level of glutamic acid in EAPPE (67.1 mg·g^−1^ protein) is similar to that reported for mealworm (65.7–60.4 mg·g^−1^ protein) [92], but lower than that of chickpea (83.3 mg·g^−1^ protein) [87]. Glutamic acid, particularly its sodium salt, contributes to the umami flavour in food. Taurine (5.2 mg·g^−1^ protein) was found in EAPPE in the present study. Taurine is nutritionally an essential amino acid for children and a conditionally essential amino acid for adults [93]. Taurine is water soluble, chemically stable at physiological pH, and abundant in animal-derived food [94]. Taurine is synthesised from cysteine, a product of methionine catabolism. In humans, the rate of taurine synthesis is exceedingly low compared with rats, livestock, and poultry [95]. Taurine acts as an antioxidant, an anti-inflammatory, and an anti-apoptotic factor in the body. Taurine is reported to exert beneficial effects on the cardiovascular, immune, muscular, reproductive, and endocrine systems [94].

The essential amino acid index (EAAI) compares these amino acids to a protein reference or to the dietary needs of humans in order to evaluate the nutritional quality of protein. The EAAI for EAPPE was determined to be 1.36 in the present study (Table 3). The EAAI values were lower than *L. rubellus* (2.12) [12], chickpea (3.5) [87], beef (3.0) [96], and mealworm (1.6) [92]. EAAI values of 0.9 or above are regarded as “good-quality proteins,” 0.8 are regarded as “useful proteins,” and 0.7 or below are regarded as “incomplete proteins.” [97]. Hence, EAPPE may be categorised as a high-quality protein. In order to have a protein intake that is balanced, an optimal EAAI and a balanced amino acid intake are required; dietitians advise including protein-containing material from multiple sources in meals. The EAAI is recognised to be an index of biological value when used to assess a single protein source [97]. The EAAI is a complete nutritional index for evaluating the quality of protein in various dietary sources because, in contrast with the protein digestibility-corrected amino acid score (PDCAAS), it takes into consideration all essential amino acids. The EAAI is considered to be an acceptable index for evaluating a protein-containing material prior to doing either feeding experiments or digestibility testing [97].

### 3.7. Functionality of EAPPE

In the following section, the functionality of EAPPE was compared with that of edible insects, which, like earthworms, are a type of invertebrate. This comparison is useful as to the best of our knowledge, the present study is the first to report the chemical and structural characteristics of *E. andrei* protein extract. 

#### 3.7.1. Colour of EAPPE

Colour is considered to be an innate indicator of whether or not unconventional foods are appealing for consumption [98]. Cayot et al. [99] described colour as the most discriminating descriptor in a sensory evaluation (*n* = 20) of Arepas fortified with either delipidated or non-delipidated *E. foetida* powder. In the present study, the EAPPE was a brown colour (Figure 2A). On the *L*, a*,* and *b** scale, EAPPE powder had a relatively low lightness of 24.04 with slight red and yellow values of 2.77 and 8.57, respectively (Table 4); this was comparatively darker (Figure 2A) than that of non-delipidated black soldier fly (*H. illucens*), which is reported to exhibit an *L** of 53.69, *a** of 4.46, and *b** of 13.08 [100]. Freeze-dried *T. molitor* larvae (mealworm) powder is reported to exhibit a much lighter colour (*L*, a*,* and *b** of 74.52, 4.73, and 8.40, respectively) [101], and the L* value was 83.36 before freeze drying. Defatting of *Protaetia beritarsis* with methanol is reported to decrease the *L** value from 18.89 to 15.39 [102]. This effect is likely to be a result of extraction of melanin pigments which are considered to be largely responsible for colour in insect material. The chroma and hue of the EAPPE were found to be 3.37 and 72.1, respectively (Table 4), in the present study. The browning index of EAPPE was determined to be 51.8, which is similar to the BI reported for *H. illucens* defatted flour (52.8) but higher than that reported for mealworm defatted flour (37.0) [103]. Interestingly, Bußler et al. [103] reported the preparation of a high-protein mealworm flour that had a browning index of 19.0, whilst the conjugate low-protein flour had a BI of 53.7, similar to that determined for EAPPE in the present study. The authors reported that over a pH range of 8–12, the mealworm protein preparation exhibited a dark brown colour [102]. The dark brown colour exhibited by the EAPPE in the present study was likely the result of the alkaline solubilization used to obtain the EAP protein extract. 

#### 3.7.2. Surface Hydrophobicity of EAPPE

The surface hydrophobicity (H_0_) value of EAPPE was found to be 1061.0 (Table 4) in the present study, which is a lot higher compared with that of *T. molitor* meal (106.25) [103]. EAPPE also had a higher H_0_ than that reported for defatted *Gryllus bimaculatus* protein concentrates (290.1 to 325) [104] and that reported for *Prionoplus reticularis* (36.3) [105]. Hydrophobic amino acids present in EAPPE, such as leucine (81.1 mg·g^−1^), isoleucine (45.6 mg·g^−1^), phenylalanine (44.0 mg·g^−1^), and valine (45.1 mg·g^−1^) likely contribute to the determined high surface hydrophobicity. Elevated H_0_ can also be induced through the denaturation of proteins during processing. Denaturation can result in the unfolding of proteins and exposure of hydrophobic regions, giving rise to a greater surface hydrophobicity [104]. Protein denaturation could arise from defatting with organic solvents, freeze drying, or alkali extraction to obtain the protein extract.

#### 3.7.3. Particle Size Distribution of EAPPE 

The particle size of the EAPPE was determined using a light scattering method (Section 2.9) and the results obtained were expressed as the hydrodynamic diameter (d_h_). The average particle size was determined to be 296 nm, and the polydispersity index (PdI), which quantifies the homogeneity, was determined to be 0.39 for EAPPE. As the PdI was greater than 0.1, the system is considered to have a polydisperse distribution of particles [106]. Similar methods of alkaline protein extraction were used by Queiroz et al. [27] and Santiago et al. [107] with *H. illucens* and *Gryllus assimilis*, respectively. The average particle size of the *H. illucens* protein extract was 174.76 nm, with a PdI of 0.262 [107], whilst the particle size of the *G. assimilis* protein isolate was 20.62 nm [27]. Both studies found that the use of a heat treatment increased the average particle size. Protein denaturation might be a probable cause for the greater average particle size of the EAPPE, as the unfolding of proteins and exposure of hydrophobic regions can result in protein aggregation and increased average particle size [104].

#### 3.7.4. Structural Characterisation of EAPPE by FTIR and DSC 

When analysing a protein extract with FT-IR, typically, the largest signal peaks principally report on the peptide amide bonds. In the present study, with FTIR analysis of EAPPE, the 2 peaks with the highest absorbance were the amide I (1638 cm^−1^) and amide II (1516 cm^−1^) peaks, with amide I corresponding to stretching of C=O and C—N bonds, whilst amide II corresponds to N—H bending and C—N stretching vibrations [108]. The amide III bands at ~1233 cm^−1^ is a considerably smaller peak (Figure 2). However, Queiroz et al. [27] associated the peak with C—N stretching vibrations and N—H deformation from amide linkages. The locations of the amide I peak (1638 cm^−1^), amide II peak (1516 cm^−1^), and amide III peaks (1232 cm^−1^) indicate the presence of β-sheet conformers in the protein extract [109]. However, the amide I peak is also very close to 1640 cm^−1^, which Fevzioglu et al. [110] associated with α-helix and random coil conformations. During the characterization of *H. illucens* protein extract by Queiroz et al. [27], the authors attributed the amide A peak (found at 3270 cm^−1^ in the present study) to N—H stretching and amide B (a split peak between ~2995–2800 cm^−1^ found in the present study) to C—H stretching. The authors also attributed the splitting of the peak to both asymmetric and symmetric C—H stretching [27].

During the thermal analysis of EAPPE using DSC, three clear endothermic peaks were generated (Figure 3). The 2 smaller peaks were found at ~167 °C and ~170 °C, with a change in enthalpy of 2.01 J/g and 1.26 J/g, respectively. A much larger peak was observed (Figure 3) at ~202 °C, with a change in enthalpy of 32.45 J/g. Similar peaks have been reported with *P. reticularis* [105] and *H. illucens* [27] protein extracts. 

When proteins are subjected to elevated temperatures, the heating energy can provide sufficient activation energy to break chemical bonds supporting the protein structure and conformation. This can cause protein denaturation, resulting in proteins losing their natural structure and function [111]. However, the 2 smaller DSC peaks at ~167 °C and ~170 °C with lower enthalpy values for EAPPE have not been observed in previous studies [27,105], but they might indicate chemical bond disruption or melting of complex carbohydrates [112], which are abundant in earthworm slime and epidermal mucus [113], and, in the form of glycoproteins [114], which might have been co-extracted when preparing the EAPPE protein extract. 

Amide A was attributed to the unfolding and denaturation of proteins, whilst the larger peak at around 200 °C was attributed to a solid melting point. *H. illucens* had a slightly lower denaturation point at 150 °C. However, the melting point was very similar to that of EAPPE (~200 °C) (Figure 3). During the analysis of freeze-dried *E. foetida* protein powder by Bou-Maroun et al. [31], DSC analysis revealed an exothermic peak attributed to protein denaturation at 42 °C. The difference in the denaturation temperature between EAPPE and the other worm species, such as *E. foetida,* might be due to differences in the composition and structure of their proteins. In general, there is a lack of research that focuses specifically on the use of DSC to study protein denaturation in earthworms. Therefore, further studies are required to improve our understanding in this area.

### 3.8. Functionality of EAPPE 

#### 3.8.1. Protein Solubility Profile of EAPPE

The present study tested EAPPE solubility over a pH range of 3 and 10. A total of 2 solubility maxima were obtained, one at pH 3 (16.5%) and the other at pH 10 (19.9%), with the lowest solubility evident between pH 5–7 (9.2–9.6%) near the isoelectric point (pI) of the EAPPE (pH ~5) (Figure 4). A similar solubility profile has been reported for *E. foetida* powder with a minimum of ~10–15% between pH 6–7 [99]. The solubility is lowest near the pI as the net charge on proteins is minimal, and their water of hydration shell is less stable, leading to a protein aggregation [115]. *T. molitor*, *G. sigillatus*, and *S. gregaria* protein extracts are also reported to exhibit pI and minimum solubility near pH 5 [113] but exhibit solubility maxima of 90–97%, which is substantially higher than that of EAPPE.

The apparent lower solubility of EAPPE could be attributed to the amino acid composition. However, EAPPE also contains a substantial amount of amino acids that have a positive impact on solubility, such as aspartic acid (88.5 mg·g^−1^), glutamic acid (67.1 mg·g^−1^), lysine (71.2 mg·g^−1^), and arginine (58.5 mg·g^−1^) [116,117]. Hence, the relatively lower apparent solubility of EAPPE is likely due to protein denaturation and hydrophobic region exposure, promoting protein aggregation [115].

#### 3.8.2. Foaming and Emulsifying Capacity and Stability of EAPPE

The foaming capacity of proteins and foam stability depends on proteins diffusing into an air–water interface and forming a film at the interface. Electrostatic and hydrophobic interactions between proteins are key in developing a stable film at an air–water interface, leading to a stable foam [118]. In the present study, EAPPE developed a stable foam (96.6% foam stability) when measured after 120 min (Figure 5A). According to Kim et al. [1], the foam stability of 3 freeze-dried defatted water-soluble protein powders from *T. molitor*, *Allomyrina dichotoma*, and *Proteatia brevitarsis seulensis* were 17%, 0%, and 30%, respectively, when measured after 60 min. The authors did mention that chitin in their samples may have destabilised the foam stability. In the present study, EAPPE was found to have a higher amount of hydrophobic amino acids and a higher surface hydrophobicity, which is likely why EAPPE exhibits superior foam stability. Mishyna et al. [29] found the foam stability of alkaline extracted protein from *A. melifera* (44.7%) and *S. gregaria* (74.1%) to be closer to that of the EAPPE (96.6%) after 120 min, whilst the foam stability of whey proteins was found to be only 14.4%. 

The emulsion capacity of EAPPE was found to be 88.8% (Table 4). This is similar to that reported for *T. molitor* (~74%), *A. dichotoma* (~72%), and *P. brevitarsis seulensis* (~85%) [116]. The slightly higher emulsion capacity exhibited by EAPPE is likely to be due to the EAPPE containing a greater proportion of hydrophobic amino acids when compared with the insect materials previously studied [116]. Zielińska [119] explained that the presence of hydrophobic amino acids and the exposure of these amino acids after denaturation facilitates interaction with lipids and enhanced emulsion activity. Moreover, EAPPE showed excellent emulsion stability (Figure 5B) that remained at 97.65% after 120 min. The emulsion stability of *T. molitor, A. dichotoma*, and *P. brevitarsis seulensis* protein extracts was tested using the same method, and the emulsion stability was reported to be ~84%, ~77%, and ~68%, respectively, after a duration of 120 min [116]. In the present study, the greater emulsion stability exhibited by EAPPE can likely be attributed to the high surface hydrophobicity, enabling greater adsorption capacity and reduced interfacial tension [1].

#### 3.8.3. Water- and Oil-Holding Capacity of EAPPE

Water-holding capacity (WHC) and oil-holding capacity (OHC) are essential techno-functional aspects of food proteins. The OHC of a food ingredient is associated with flavour release, mouthfeel, and palatability, whilst WHC is critical for understanding the functional impact of a product in terms of visual acceptability and food texture [120]. The WHC and OHC are defined by the volume (mL) of water or oil that is retained per gram of material. In the present study, EAPPE was found to have a WHC of 2.93 mL·g^−1^ and a slightly higher OHC of 4.27 mL·g^−1^. *G. assimilis, Acheta domesticus,* and *Zophobas morio* protein extracts have been reported to exhibit much higher WHC (~4.40 g/g, 5.86 g/g, and ~5.18 g/g, respectively), compared with their OHC (3.10 g/g, ~3.02 g/g, and 3.73 g/g, respectively) [119]. In the present study with EAPPE, the opposite was found. This is likely linked to the relatively high proportion of hydrophobic amino acids in EAPPE. Protein conformation and surface hydrophobicity affect oil holding capacity, which relies on protein entrapping oil through capillary interactions [120].

#### 3.8.4. Coagulation of EAPPE

Heat coagulation is an important property for gelling of proteins, and higher coagulation results in firmer gels [121]. In the present study, the heat coagulation of EAPPE at the pI of pH 5.0 was 4.21%, whilst the heat coagulation at pH 7.0 was 12.6% (Table 4). These results are much lower than those reported for alkaline-extracted *A. melifera,* which exhibited heat coagulation at pI of 18.2% and at pH 7 of 30.2% [29]. The lower coagulation of protein in EAPPE can be attributed to the higher surface hydrophobicity of EAPPE, which in turn would have a negative impact on solubility and heat coagulation [29]. Both the present study of EAPPE and that of alkaline-extracted *A. melifera* [27] report greater heat coagulation at pH 7 compared with their respective pI. This is associated with higher solubility as a result of greater electrostatic repulsion between proteins due to amino acid ionized side chains stabilizing water of hydration around protein molecules, resulting in less aggregation [29]. 

## 4. Conclusions and Future Prospects

The current study offers a substantial amount of original information on the proximate analysis, fatty acid composition, mineral profile, protein extraction, amino acid composition, and the chemical and structural features of the protein extract derived from *E. andrei* earthworm species that was wild-harvested in New Zealand.

Many earthworm species have not been studied extensively, including *E. andrei*. In the present study, *E. andrei* poeder (EAP) was found to have substantial nutrient content, with a protein content of 53.7% DW and fat content of 19.3% DW. The most abundant SFA, MUFA, and PUFA were found to be lauric acid, myristoleic acid, and linoleic acid, respectively. The lipid composition contains high levels of UFA compared with SFA. However, the moderate level of SFA and its relation to consumption is a consideration with EAP as a food. The lipid nutritional indices (IT and ω-6/ω-3) of EAP indicate that EAP is a healthy material. However, the HPI, IA, and H/H values determined in the present study are lower for EAP compared with that of other species of earthworms, suggesting there may be some variation due to differences in species, habitat, and diet.

ICP-MS analysis detected 15 minerals in EAP with relatively high levels of calcium, potassium, phosphorus, sodium, iron, zinc, and manganese. However, levels of some heavy metals (cadmium, arsenic, and lead) were detected, which presents food consumption and safety considerations.

The yield of protein obtained in *E. andrei* protein extracts (EAPPE) and protein analysis by FTIR indicate that EAPPE is a protein-rich material. From a nutritional perspective, the total amino acid content (373.3 mg·g^−1^) and EAAI (1.4) of EAPPE indicate that consumption of EAPPE can meet daily nutritional requirements in human adults. This study also provides novel information on the chemical and structural protein functionalities. EAPPE was found to exhibit substantial foaming and emulsion properties, which can be valuable in food technology formulations, such as ice cream, dough, and salad dressing.

The present study, therefore, provides information important to the utilisation of earthworm material, such as *E. andrei* in powder form, in food and feed formulations, along with the use of *E. andrei* protein extracts as an ingredient in other food applications. In addition, the present study is the first comprehensive report on the techno-functional and nutritional properties of *E. andrei* earthworm powder and protein extract, indicating their potential for use in the food and feed industry. However, further studies are needed that investigate the farming of *E. andrei* with respect to controlling the growth environment and rearing conditions to minimise heavy metal accumulation. Once appropriate rearing techniques have been established to reduce heavy metal accumulation, EAP and EAPPE could be further subjected to human sensory evaluation and evaluate consumer acceptability.

## Figures and Tables

**Figure 1 foods-12-01948-f001:**
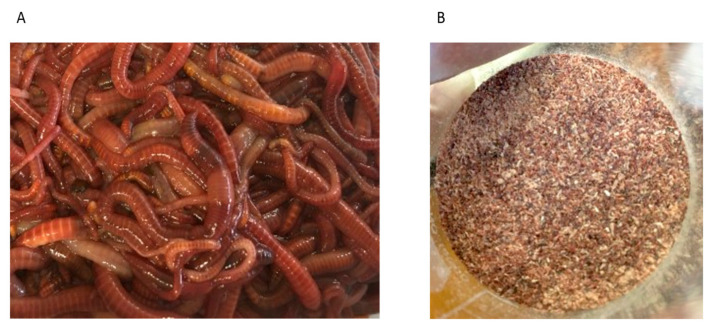
(**A**) *E. andrei* before drying and (**B**) dried powdered *E. andrei*.

**Figure 2 foods-12-01948-f002:**
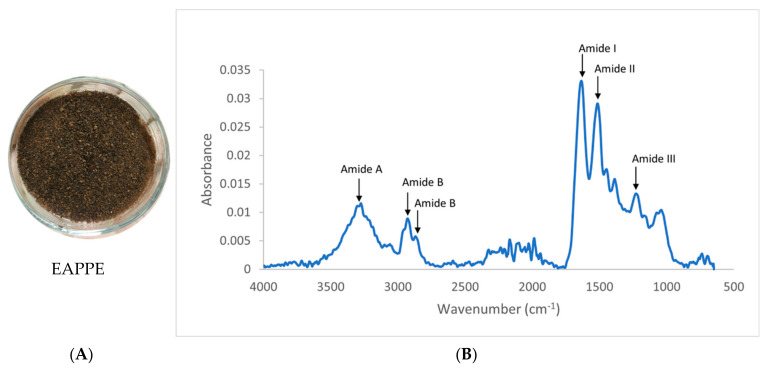
Image of EAPPE (**A**) and FTIR spectra of the EAPPE (**B**).

**Figure 3 foods-12-01948-f003:**
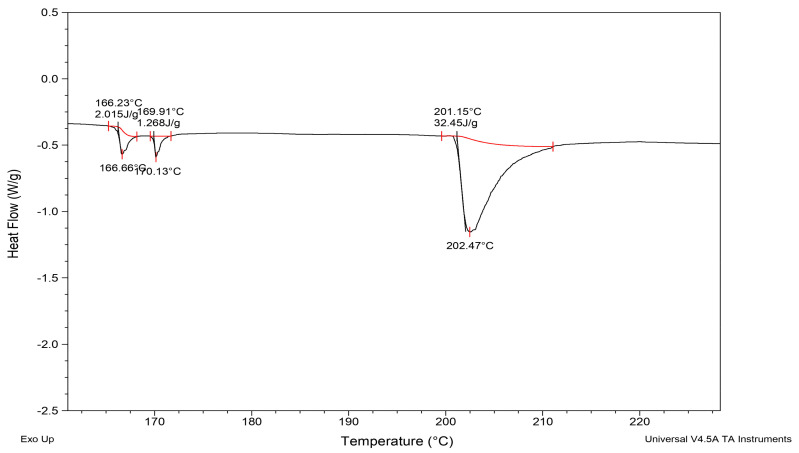
Differential scanning calorimetry (DSC) thermograms of EAPPE.

**Figure 4 foods-12-01948-f004:**
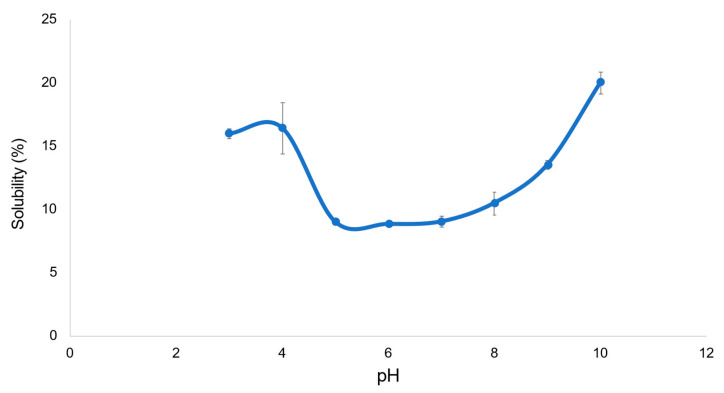
The solubility of EAPPE at various pH.

**Figure 5 foods-12-01948-f005:**
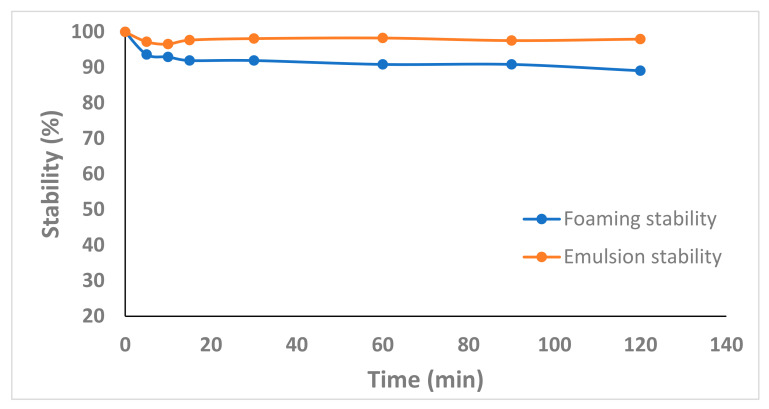
The foam and emulsion stabilities of EAPPE.

**Table 1 foods-12-01948-t001:** Mean content fatty acids (% FA, g FA/100 g of dried EAP material).

Fatty Acid		% Total Fatty Acid	g FA/100 g Dried EAP
Undecanoic acid	C11:0	0.54 ± 0.1	0.13 ± 0.02
Lauric acid	C12:0	20.3 ± 3.3	4.76 ± 0.57
Myristic acid	C13:0	1.05 ± 0.2	0.25 ± 0.04
Palmitic acid	C14:0	12.8 ± 1.4	3.01 ± 0.20
Margaric acid	C17:0	2.18 ± 0.8	0.51 ± 0.05
Stearic acid	C18:0	5.03 ± 0.8	1.18 ± 0.05
Arachidic acid	C20:0	2.57 ± 0.3	0.30 ± 0.14
Total Saturated Fatty Acid	SFA	44.4 ± 6.1	10.14 ± 1.08
Myristoleic acid	C14:1	11.2 ± 0.7	2.64 ± 0.13
Pentadecanoic acid	C15:1	1.09 ± 0.3	0.26 ± 0.18
Palmitoleic acid (cis)	C16:1 cω7	6.94 ± 1.5	1.03 ± 0.76
Heptadecenoic acid	C17:1	1.62 ± 0.4	0.39 ± 0.05
Oleic acid (cis)	C18:1 cω9	4.19 ± 2.9	1.01 ± 0.02
Erucic acid	C21:0	1.49 ± 0.6	0.38 ± 0.05
Total Monounsaturated Fatty Acid	MUFA	26.5 ± 4.1	5.69 ± 1.15
Linoleic acid	(cis)C18:2 ω6	7.96 ± 0.2	1.88 ± 0.05
α-Linolenic acid	C18:3 ω3	2.03 ± 0.5	0.14 ± 0.02
Dihomo-gamma-linolenic acid	C20:3 ω6	5.70 ± 0.5	0.21 ± 0.08
Arachidonic acid	C20:4 ω6	6.39 ± 0.5	0.83 ± 0.04
Eicosapentaenoic acid (EPA)	C20:5 ω3	6.29 ± 0.6	0.93 ± 0.08
Total Polyunsaturated Fatty Acid	PUFA	28.4 ± 0.2	3.99 ± 0.27
Polyunsaturated fatty acid/saturated fatty acid	PUFA/SFA	0.23 ± 0.03	—
Omega 6: Omega 3	ω6: ω3	4.11 ± 1.1	—
Index of atherogenicity	IA	1.32 ± 0.3	—
Index of thrombogenicity	IT	0.55 ± 0.1	—
Hypocholesterolemic/ hypercholesterolemic ratio	H/H	0.44 ± 0.2	—
Health-promoting index	HPI	0.78 ± 0.2	—

All values are expressed as mean ± Standard Deviation (SD); *n* = 3; ‘—’ not determined.

**Table 2 foods-12-01948-t002:** Mineral content of EAP macrominerals (a), microminerals (b), heavy metals, and (c) in mg·kg^−1^ DW.

(a) Macrominerals						
	Sodium	Magnesium	Phosphorus	Potassium	Calcium	
*E. andrei*	4056.7 ± 257.9	744.7 ± 31.8	6600.0 ± 225.2	8220.0 ± 264.6	2396.7 ± 205.0	
^1^ DORM-4	14,052 ± 1006	897 ± 76	7113 ± 497	13,352 ± 962	2391 ± 181	
Recovery (%)	100	99	89	86	101	
^2^ Limit of detection	4	2	10	4	8	
^3^ Limit of quantification	12	6	30	12	24	
**(b) Microminerals**						
	**Chromium**	**Copper**	**Iron**	**Zinc**	**Selenium**	**Manganese**
*E. andrei*	0.2 ± 0.0	8.0 ± 0.4	244.7 ± 20.5	84.7 ± 1.8	0.6 ± 0.1	25.6 ± 0.6
^1^ DORM-4	1.75 ± 0.2	15 ± 0.9	333 ± 0.9	51 ± 4.8	3.2 ± 0.5	3.04 ± 0.2
Recovery (%)	94	97	97	98	93	96
^2^ Limit of detection	0.02	0.06	0.8	2	0.02	0.02
^3^ Limit of quantification	0.06	0.2	2	6	0.06	0.06
**(c) Heavy metals**						
	**Arsenic**	**Vanadium**	**Lead**	**Cadmium**		
*E. andrei*	2.3 ± 0.7	0.2 ± 0.1	0.2 ± 0.0	2.2 ± 0.2		
^1^ DORM-4	6.4 ± 0.33	1.52 ± 0.09	0.34 ± 0.04	0.30 ± 0.02		
Recovery (%)	93	97	85	100		
^2^ Limit of detection	0.010	0.006	-	-		
^3^ Limit of quantification	0.06	0.02	0.03	0.03		

Triplicate (*n* = 3) analysis. BDL—below detectable level. ^1^ DORM-4—fish protein certified reference material (mg·kg^−1^ DW). ^2^ Limit of detection = the lowest amount that can be detected (confidence of 99%) using ICP-MS. ^3^ Limit of quantitation = lowest concentration of the substance that can be determined to establish accuracy, precision, and replicability.

**Table 3 foods-12-01948-t003:** Amino acid composition (mg·g^−1^ protein) of EAPPE and the required daily essential amino acids for adults.

Amino Acid	EAPPE	^2^ Adult Daily Requirement 1985 FAO/WHO/UN
**Essential amino acids**		
Histidine (His)	27.9 ± 2.7	15
Isoleucine (Ile)	45.6 ± 3.2	30
Phenylalanine (Phe)	44.0 ± 3.3	30
Threonine (Thr)	38.9 ± 3.9	23
Lysine (Lys)	71.2 ± 8.1	45
Leucine (Leu)	81.1 ± 6.3	59
Valine (Val)	45.1 ± 3.2	39
Methionine (Met)	19.6 ± 1.6	22 (Met + Cys)
Total essential amino acid (TEAA)	373.3 ± 9.1	263
^3^ EAAI	1.4 ± 0.05	
**Non-essential amino acids**		
Aspartic acid (Asp)	88.5 ± 5.5	
Cysteine (Cys)	12.6 ± 1.2	
Glycine (Gly)	34.8 ± 3.0	
Arginine (Arg)	58.5 ± 6.3	
Alanine (Ala)	47.6 ± 4.2	
Serine (Ser)	39.5 ± 3.0	
Tyrosine (Tyr)	39.3 ± 3.1	
Glutamic acid (Glu)	67.1 ± 5.0	
Proline (Pro)	37.2 ± 1.9	
^1^ Taurine (Tau)	5.2 ± 0.4	
Asparagine (Asn)	2.6 ± 0.3	
Sum of total amino acids	806.1 ± 80.6	

^1^ Tau = taurine. Tryptophan was not determined in this study. ^2^ Adult daily requirement for amino acids obtained from the 1985 FAO/WHO/UN [26]. ^3^ EAAI = essential amino acid index. Mean ± standard deviation (SD); *n* = 3.

**Table 4 foods-12-01948-t004:** Techno-functionality and physiochemical characteristics of the EAPPE suspensions.

	EAPPE
Colour *L**	24.04 ± 0.4
*a**	2.77 ± 0.2
*b**	8.57 ± 0.3
C	3.37 ± 0.06
h^°^	72.1 ± 0.8
BI (browning index)	51.8 ± 3.0
Water-holding capacity (WHC) (g·mL^−1^)	2.93 ± 0.9
Oil-holding capacity (OHC) (g·mL^−1^)	4.27 ± 0.5
Surface hydrophobicity	1061.0 ± 1.7
The polydispersity index (PDI)	0.39 ± 0.02
Hydrodynamic diameter (d_h_, nm)	296 ± 24.1
Emulsion capacity (%)	88.8 ± 4.8
Heat coagulation at pH 5 (%)	4.21 ± 0.2
Heat coagulation at pH 7 (%)	12.6 ± 3.8

All values are expressed as mean ± standard deviation (SD); *n* = 3.

## Data Availability

Data is contained within the article.

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
