# Peer review of "Earth Worming—An Evaluation of Earthworm (Eisenia andrei) as an Alternative Food Source"

_foods, 2023, doi:10.3390/foods12101948_

Round 1
Reviewer 1 Report
The manyuscript is well written and have good sceitific merit. The parameters used for deriving the results and conclusion are novel and exhaustive. The hypothesis is well explianed and sound. The language is clear and easy to understand.
I have following suggestion:
i. Abstract: please mention the source/ place of the Earthworms, so to provide better clarity; as authors mention the area specific in last para of intro for ref 19.
ii. Keyword: protein and fat comes under proximate, so may be deleted
iii. Para 2: Introductio carbs may be written as carbohydrates
iv. Ref 19 in last para of intro- please write place
v. Section 2.1: please check whether any ethical guidelines required?
vi. Section 2.18: please add more descriptions
vii. Table 1 seems too short, may be written in text or merged with other table.
viii. Please check the reference citation in text, I think after name no need for , sigh as Lastname, et al. may be Lastname et al.
ix. Results and discussion: well written and well supported by relevant literatures.
x. Conclusion: may be concised or add conclusion and future prospects etc to justify it
Language is clear and easy to understand.
Author Response
The authors would like to thank the reviewers for their comments on our work.
Reviewer 1
Comments and Suggestions for Authors
The manuscript is well written and have good scientific merit. The parameters used for deriving the results and conclusion are novel and exhaustive. The hypothesis is well explained and sound. The language is clear and easy to understand.
We thank the Reviewer for their positive comment.
I have following suggestion:
- Abstract: please mention the source/ place of the Earthworms, so to provide better clarity - as authors mention the area specific in last para of intro for ref 19.
Done.
- Keyword: protein and fat comes under proximate, so may be deleted.
Done.
iii. Para 2: Introduction carbs may be written as carbohydrates.
Done.
- Ref 19 in last para of intro- please write place.
Done.
- Section 2.1: please check whether any ethical guidelines required?
No ethics are required at the University of Otago for research on earthworms.
- Section 2.18: please add more descriptions.
Section 2.18 has been re-written.
vii. Table 1 seems too short, may be written in text or merged with other table.
On reconsideration, this table has been removed and the data are provided in the text (Section 3.1)
viii. Please check the reference citation in text, I think after name no need for , sigh as Lastname, et al. may be Lastname et al.
The MDPI journal guideline requires a comma after the author’s name, so not changes were made.
- Results and discussion: well written and well supported by relevant literatures.
We thank the Reviewer for their positive comment.
- Conclusion: may be concise or add conclusion and future prospects etc to justify it
The heading for section 4 has been modified to include ‘future prospects’.
Reviewer 2 Report
The study is interesting and the experiment appears to be well conducted. However, there is a lack of application test. As a minimum, authors should incorporate the protein extract or worm powder into some sort of food to test the efficacy and sensory properties. This additional experiment is important and required, otherwise the research cannot be justified.
For data interpretation, the exceedingly high temperatures of "melting" around 170 (2 peaks) cannot be explained as protein structure unfolding. They are most likely due to chemical bond breaking beyond structural change and denaturation.
Author Response
The authors would like to thank the reviewers for their comments on our work.
Reviewer 2
Comments and Suggestions for Authors
The study is interesting and the experiment appears to be well conducted.
We thank the Reviewer for their positive comment.
However, there is a lack of application test. As a minimum, authors should incorporate the protein extract or worm powder into some sort of food to test the efficacy and sensory properties. This additional experiment is important and required, otherwise the research cannot be justified.
We understand the Reviewer’s comment, however, (i) the present study comprises a substantial manuscript of nearly 30 pages, (ii) the scope of the study was to provide a comprehensive fundamental evaluation of farmed earthworms, and (iii) as alluded to in the manuscript, the authors have some concerns in relation to human sensory evaluation, due to ethical considerations in relation to the heavy metal concentration found in the present study.
For data interpretation, the exceedingly high temperatures of "melting" around 170 (2 peaks) cannot be explained as protein structure unfolding. They are most likely due to chemical bond breaking beyond structural change and denaturation.
The text has been reconsidered and modified accordingly.
Round 2
Reviewer 2 Report
The usefulness of the study is determined by its applicability of worm protein as food or food ingredient. Therefore, despite the concern for heavy metals, etc. (which seems to diminish the value of the study?), an experiment to show consumers' perception of the worm protein and products must be included. Otherwise, as indicated in the prior review, the present study does not contribute to new food production -- there have been many other published studies on earn worm protein. Authors need to justify their present work; the inclusion of sensory evaluation (e.g., sniffing? color? appearance) would make the work "new".
Author Response
Reviewer comment: “The usefulness of the study is determined by its applicability of worm protein as food or food ingredient. Therefore, despite the concern for heavy metals, etc. (which seems to diminish the value of the study?), an experiment to show consumers’ perception of the worm protein and products must be included. Otherwise, as indicated in the prior review, the present study does not contribute to new food production -- there have been many other published studies on earn worm protein. Authors need to justify their present work; the inclusion of sensory evaluation (e.g., sniffing? color? appearance) would make the work “new”.”
Response: We agree with the Reviewer in part about their comment that “the usefulness of the study is determined by its applicability of worm protein as food or food ingredient”. Ultimately, the end goal of any food science-related article proposing a ‘novel’ ingredient is whether that ingredient is edible. (We maintain, though, that scientific advancement is not always utilitarian, and that science can also be done for mere intellectual curiosity). But we do not agree that this means the entire gamut of experiments (i.e., nutrition characterisation, functionality assessment, and sensory evaluation) must be in one article. Our research group has an active program investigating ‘novel’ foods like edible insects and earthworms, and sensory assessment of these food materials is something we plan to do, once we have a clear understanding of the risks associated with the material. However, the scope of the present study was to “provide a comprehensive evaluation of nutritional composition (proximate analysis and profiles of fatty acids and minerals) and techno-functional properties (foaming and emulsion stability and capacity) of earthworm (Eisenia andrei) powder (EAP), sourced in New Zealand”, rather than do a sensory evaluation of the material. We maintain that the compositional analysis is an important precondition to any follow-up work on sensory evaluation. In any case, our work has shown that these earthworms are high in heavy metals. This finding, contrary to the point of the Reviewer, in no way diminishes the value of the study, for how would one know that these earthworms could be toxic if we hadn’t reported it? That heavy metals can bioaccumulate in an organism that lives in the soil is no surprise and gives insights into overcoming this challenge. This is why we have stated in the conclusion that changes in rearing conditions could be used to alter the mineral profile of the earthworm into forms that are safe to consume. Until such rearing experiments are done (which will require time), we cannot, as ethical scientists, do a sensory evaluation on these earthworms. We hope that the Reviewer understands this point.
We are also aware of the published literature on the earthworm (see the list below). As seen from the list, none of these studies combined composition analysis with sensory evaluation. Another thing to note is that almost none of these studies looked at the techno-functional properties of the earthworm materials (which we have in our manuscript). And there is only one published study (#7 below) on the earthworm species (Eisenia andrei), which we used in our research. The lack of studies on this species is another justification for our work.
- Garczyńska, M.; Kostecka, J.; Pączka, G.; Mazur-Pączka, A.; Cebulak, T.; Butt, K.R. Chemical Composition of Earthworm (Dendrobaena veneta Rosa) Biomass Is Suitable as an Alternative Protein Source. International Journal of Environmental Research and Public Health, 2023, 20, 3108.
- Parolini, M.; Ganzaroli, A.; Bacenetti, J. Earthworm as an alternative protein source in poultry and fish farming: current applications and future perspectives. Science of the Total Environment 2020, 734, 139460.
- Ding, S.; Lin, X.; He, S. Earthworms: A source of protein. Journal of Food Science Engineering 2019, 9, 159-170.
- Rahayu, R.; Hudha, A.M.; Permana, F.H. Analysis of Nutritional Content of Fresh Sea Worm Honingka (Siphonosoma australe-australe) as a Potential Food Source for Communities. In Proceedings of the IOP Conference Series: Earth and Environmental Science, 2019; p. 012026.
- Sun, Z.; Jiang, H. Nutritive evaluation of earthworms as human food. Future Foods 2017, 37.
- Bahadori, Z.; Esmaielzadeh, L.; Karimi-Torshizi, M.; Seidavi, A.; Olivares, J.; Rojas, S.; Salem, A.; Khusro, A.; López, S. The effect of earthworm (Eisenia foetida) meal with vermi-humus on growth performance, hematology, immunity, intestinal microbiota, carcass characteristics, and meat quality of broiler chickens. Livestock Science 2017, 202, 74-81.
- Rodrigues, M.; Carlesso, W.M.; Kuhn, D.; Altmayer, T.; Martini, M.C.; Tamiosso, C.D.; Mallmann, C.A.; De Souza, C.F.V.; Ethur, E.M.; Hoehne, L. Enzymatic hydrolysis of the Eisenia andrei earthworm: Characterisation and evaluation of its properties. Biocatalysis and Biotransformation 2017, 35, 110-119.
- Ukoha, O.; Onunkwo, D.; Obike, O.; Nze, U. Proximate, vitamin and mineral composition of earthworm (Hyperiodrilus euryaulos) cultured indifferent Animal dung Media. Nigerian Journal of Animal Production 2017, 44, 257-261.
- Gunya, B.; Masika, P.J.; Hugo, A.; Muchenje, V. Nutrient composition and fatty acid profiles of oven-dried and freeze-dried earthworm Eisenia foetida. Journal of Food and Nutrition Research 2016, 4, 343-348.
- Bou-Maroun, E.; Loupiac, C.; Loison, A.; Rollin, B.; Cayot, P.; Cayot, N.; Marquez, E.; Medina, A.L. Impact of preparation process on the protein structure and on the volatile compounds in Eisenia foetida protein powders. Food and Nutrition Sciences 2013, 4, 1175.
- Lourdumary, A.B.; Uma, K. Nutritional evaluation of earthworm powder (Lampito mauritii). Journal of Applied Pharmaceutical Science 2013, 3, 082-084.
- Sinha, R.K.; Valani, D.; Chauhan, K.; Agarwal, S. Embarking on a second green revolution for sustainable agriculture by vermiculture biotechnology using earthworms: reviving the dreams of Sir Charles Darwin. Journal of Agricultural Biotechnology and Sustainable Development 2010, 2, 113.
- Li, K.; Li, P.; Li, H. Earthworms helping economy, improving ecology and protecting health. International Journal of Global Environmental Issues 2010, 10, 354-365.
- Istiqomah, L.; Sofyan, A.; Damayanti, E.; Julendra, H. Amino acid profile of earthworm and earthworm meal (Lumbricus rubellus) for animal feedstuff. Journal of the Indonesian Tropical Animal Agriculture 2009, 34, 253-257.
- Sogbesan, A.; Madu, C. Evaluation of earthworm (Hyperiodillus euryaulos, clausen, 1914; oligocheata: eudrilidae) meal as protein feedstuffs in diet for Heterobranchus longifilis valenciennes, 1840 (teleostei, clariidae) fingerlings under laboratory condition. Research Journal of Environmental Sciences 2008, 2, 23-31.
- Medina, A.; Cova, J.; Vielma, R.; Pujic, P.; Carlos, M.; Torres, J. Immunological and chemical analysis of proteins from Eisenia foetida earthworm. Food and Agricultural Immunology 2003, 15, 255-263.
- Rondón, V.; Ovalles‐Durán, J.; León‐Leal, M. Nutritional value of earthworm flour (Eisenia fetida) as a source of amino acids and its quantitative estimation through reversed phase Chromatography (HPLC) and pre‐column derivation withoff‐phthalaldehyde. ARS Pharmacy 2003, 44, 43-58.
- Gaddie, R.E.; Douglas, D.E. Earthworms for ecology and profit. Scientific Earthworm Farming, Bookworm Publishing Company 1975, 1, 175.
In addition to all the above, we have modified the last few sentences of our conclusion section, to emphasise some of the points above. The read:
The present study, therefore, provides information important to utilisation of earthworm material, such as E. andrei in powder form, in food and feed formulations, along with the use of E. andrei protein extracts as an ingredient in other food applications. The present study is the first comprehensive report on the techno-functional and nutritional properties of E. andrei earthworm powder and protein extract, indicating their potential for use in the food and feed industry. However, further studies are needed that investigate the farming of E. andrei with respect to controlling the growing environment and rearing conditions to minimise heavy metal accumulation. Once the appropriate rearing techniques have been used to modify the heavy metal contents, EAP and EAPPE could be further subjected to human sensory evaluation to understand their consumer acceptability.